



# Model analysis of biases in satellite diagnosed aerosol effect on cloud liquid water path

Harri Kokkola[1], Juha Tonttila[2], Silvia Calderón[1], Sami Romakkaniemi[1], Antti Lipponen[1], Aapo Peräkorpi[3], Tero Mielonen[1], Edward Gryspeerdt[4], Timo H. Virtanen[1], Pekka Kolmonen[1], and Antti Arola[1]

[1]Atmospheric Research Centre of Eastern Finland, Finnish Meteorological Institute, Kuopio, Finland
[2]CSC - IT Center for Science, Espoo, Finland
[3]Datrix S.p.A., Milan, Italy
[4]Grantham Institute - Climate Change and the Environment, Imperial College London, London, UK

**Correspondence:** Harri Kokkola (harri.kokkola@fmi.fi)

**Abstract.** The response in cloud water content to changes in cloud condensation nuclei remains one of the major uncertainties in determining how aerosols can perturb cloud properties. In this study, we used an ensemble of large eddy simulations of marine stratocumulus clouds to investigate the correlation between cloud liquid water path and the amount of cloud condensation nuclei. We compare this correlation directly from the model to the correlation derived using equations which are used to retrieve liquid water path from satellite observations. Our comparison shows that spatial variability in cloud properties and instrumental noise in satellite retrievals of cloud optical depth and cloud effective radii result in bias in satellite-derived liquid water path. In depth investigation of high-resolution model data shows that in large part of a cloud, the assumption of adiabaticity does not hold which results in a similar bias in LWP-CDNC relationship as seen in satellite data. In addition, our analysis shows a significant positive bias of between 18 % and 40 % in satellite-derived cloud droplet number concentration. However, for the individual ensemble members, the correlation between the cloud condensation nuclei and the mean of the liquid water path was very similar between the methods. This suggests that if cloud cases are carefully chosen for similar meteorological conditions and it is ensured that cloud condensation nuclei concentrations are well-defined, changes in liquid water can be confidently determined using satellite data.

## 1 Introduction

Clouds are in a crucial role in the Earth's climate affecting the radiative balance of the Earth as they cover majority of the Earth surface having high reflectivity on the incoming solar radiation and absorbing outgoing thermal radiation (Bellouin et al., 2020; Forster et al., 2021). As aerosol can perturb the cloud properties, accurate knowledge of how aerosol-cloud interactions affect clouds will allow for better estimation on how changes in anthropogenic emissions affect the Earth's radiative balance and thus the climate. Satellite based estimates of aerosol effects on clouds have proven to be challenging to interpret as they have not always supported the theoretical assumptions of decreasing cloud droplet sizes with increasing number of cloud droplets (Twomey, 1974; Jia et al., 2019) or the increase of cloud liquid water content with increasing number of cloud droplets





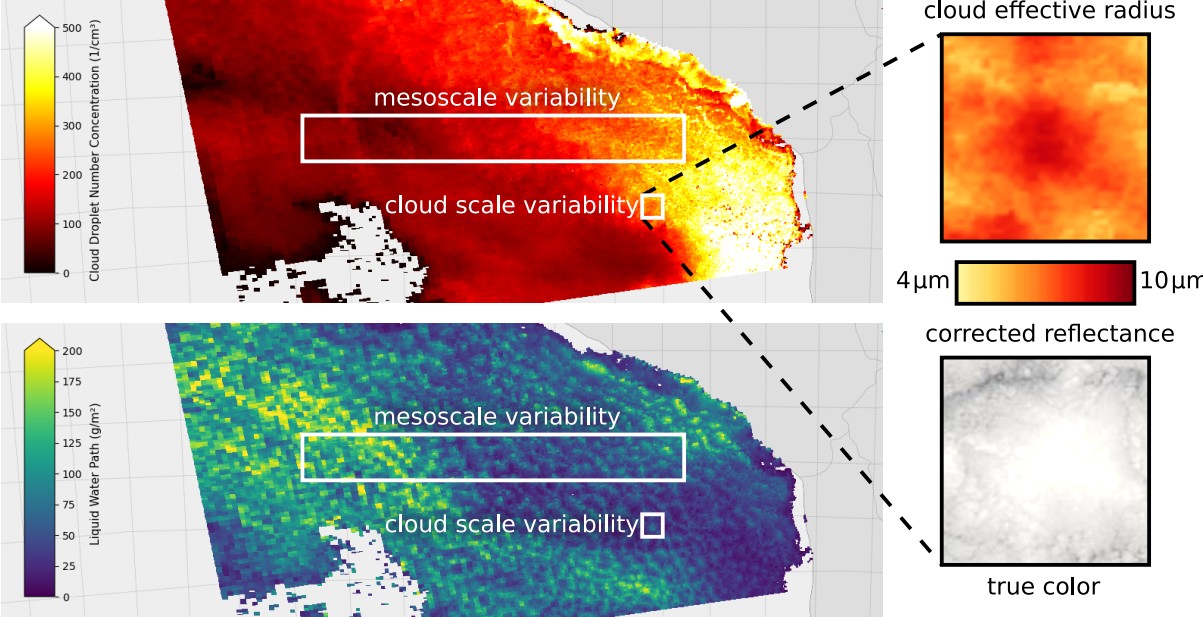

**Figure 1.** Cloud properties of a stratocumulus cloud deck west of Peru and Chile over South Pacific on Aug 30th, 2003. The upper left panel shows the calculated CDNC and lower left panel shows the retrieved LWP from Moderate Resolution Imaging Spectroradiometer (MODIS) Level-2 (L2) Collection 6.1 (Platnick et al., 2015). Right panels show a magnification of the structure of a cloud cell within the cloud field denoting the cloud effective radius and the cloud reflectance for the corresponding cloud cell.

(Albrecht, 1989; Gryspeerdt et al., 2019). These mixed results have been attributed to several counteracting physical processes (Feingold et al., 2022; Gryspeerdt et al., 2022; Glassmeier et al., 2021; Zhang et al., 2024), but also challenges in satellite retrievals (Feingold et al., 2022; Arola et al., 2022).

Arola et al. (2022) showed that variability in positively correlated cloud droplet number concentration (CDNC) and liquid water path (LWP) data will dilute the correlation and can even result in an apparent strongly negative correlation between CDNC and LWP. However, in that study, the causes of variability were not studied further. Such variability can come from 1) internal variability in clouds originating from the circulation within clouds, e.g., in updrafts at the center of the cloud cells and downdrafts at the edges of the cell, 2) mesoscale variability in meteorological conditions and phase of the cloud evolution, 3)

instrumental noise in satellite retrievals.

     Figure 1 shows the cloud properties in a stratocumulus deck west of Peru and Chile, South America. In the figure, the wide rectangles show regions in the cloud field which show mesoscale variability in CDNC and LWP while the small squares indicate internal variability in CDNC and LWP within a cloud cell. The smaller squares are magnified in the right-hand side panels, showing both the effective radius and the reflectivity of the cloud at visible wavelengths.

The wide rectangle in the figure shows a region with significant mesoscale variabilities in cloud properties. Although some of the variability can come from changes in aerosol, over that region the cloud top effective radius and liquid water path follow



the changes in cloud top pressure. The cloud top pressure increases from $850\,\mathrm{hPa}$ to $890\,\mathrm{hPa}$ going from west to east indicating higher boundary layer which might affect boundary layer dynamics and thus cloud properties.

As for the small squares in Figure 1, we can see that the retrieved cloud effective radius decreases towards the cloud cell edges. The decrease in retrieved cloud effective radius results from the entrainment mixing at the cloud top and downdrafts in the cloud cell boundaries which both reduce the liquid water content. However, calculation of CDNC based on the effective radius, and assuming constant sub-adiabaticity, would lead to increased CDNC at the cell boundaries (see Equation (2) in Section 2). In addition to actual variability in physical properties of clouds, satellite retrievals include uncertainties and instrument noise causing another potential source of bias in the satellite-derived correlation between CDNC and LWP. All these different

sources of variability are potential causes for biasing the estimate of aerosol effect on LWP as shown by Arola et al. (2022)

     In this study we use a cloud resolving large eddy simulations (LES) model to investigate the relative contribution of these sources of variability (cloud condensation nuclei, cloud structure, and noise in satellite retrievals) to the correlation between CDNC and LWP. We will analyse how the diagnosed response in liquid water path (LWP) to perturbed aerosol concentrations differ when calculated with equations used in the satellite retrievals of LWP, compared to LWP diagnosed directly from the

LES model.

## 2    METHODS

### 2.1    Model description

We simulated the effect of aerosol concentration on cloud properties, especially the liquid water content using UCLALES-SALSA large eddy simulations model (Stevens et al., 2005; Kokkola et al., 2008; Tonttila et al., 2017; Ahola et al., 2020).

In this model setup, UCLALES which simulates the dynamics of the boundary layer is coupled to the aerosol-cloud model SALSA which simulates aerosol and cloud droplet microphysics. SALSA has a sectional description for aerosol particles, cloud and precipitation droplets, and ice crystals.

     The model is built around UCLALES, a platform for idealized cloud simulations. The model resolves the turbulent flow in a three-dimensional cartesian grid with cyclic boundary conditions. The main prognostic scalar variables include the liquid

water potential temperature and tracer variables describing water vapor and liquid water mixing ratios. When coupled with the sectional aerosol-cloud microphysics model SALSA, the set of prognostic scalars is vastly extended, now including the number and mass mixing ratios for each size section of four particle categories, comprising aerosol particles, cloud droplets, drizzle/precipitation and ice (the latter not used in this study). The size resolved framework is used to describe particle growth via condensation and coalescence processes in all categories. Aerosol cloud activation is determined directly from the resolved

particle growth whereas the transition between cloud droplets and drizzle is diagnosed from the resolved collision-coalescence process. UCLALES-SALSA has been validated against observations in liquid phase clouds and fogs and found to reproduce the observed droplet distributions (e.g. Boutle et al., 2018; Calderón et al., 2022). More details on the model and simulations are given in Supplementary Information Sections S1 and S2.



### 2.1.1 Experiment setup

To better understand the features of the LWP response to changes in CDNC seen in the satellite data, UCLALES-SALSA was configured for a typical marine stratocumulus cloud setup (nocturnal drizzling stratocumulus cloud DYCOMS-II RF02 of the Second Dynamics and Chemistry of Marine Stratocumulus field campaign (Ackerman et al., 2009)), representing a very commonly occurring cloud type which allows for disentangling the potential underlying numerical biases related to satellite retrievals. The horizontal model domain size was $51 \times 51$ km with a resolution of 75 m with vertical domain extending up to 75 1.4 km with vertical resolution of 20 m. Vertical profiles of atmospheric variables used for model initialization are shown in Figure S1 in Supporting Information.

For aerosol, we used a bi-modal size distribution. The Aitken mode is centered at 0.022 μm with a standard deviation of 1.2 and total number concentration of 150 cm$^{-3}$. The accumulation mode is centered at 0.120 μm with standard deviation of 1.7. To determine the effect of aerosol on cloud properties, we ran a set of 3 simulations with cloud condensation nuclei 80 (CCN) equivalent to initial accumulation mode aerosol concentrations of 65, 150 and 300 cm$^{-3}$ (corresponding to the total number concentrations of 78, 180 and 360 cm$^{-3}$ for the whole size distribution). Size distributions are illustrated in Figure S2 in Supporting Information.

The simulations span over 14 h and the model output was sampled 2 h, 6 h, and 10 h from the start of the run. These time intervals allow us to draw samples from different cloud structures, as the microphysical properties and circulation structures 85 are allowed to evolve freely in the model. In particular, the simulations with the lowest initial aerosol concentrations exhibit a clear drizzle induced transition from closed stratocumulus cells to an open cell structure within 10 hours of model time.

It is well-known that satellite-based CDNC values are biased because radiances used in the CER retrievals correspond to an optically thicker region below cloud top. Platnick (2000) established that infrared radiance fluxes from liquid clouds include all reflected photons that penetrate to a maximum optical depth equivalent to up to 3.5 units below cloud top (Grosvenor et al., 90 2018b) depending on the viewing geometry and cloud heterogeneity (Grosvenor et al., 2018a). With CER values that are smaller than those expected at cloud top, the cloud-top based pseudo-adiabatic model inevitably fails, producing satellite-retrievals of CDNC and LWP that are different from real ones (Grosvenor et al., 2018a).

In this study, we followed a sampling methodology that mimics this so-called penetration depth bias (Grosvenor et al., 2018a). We determined the CER and CDNC values for the top part of a cloud based on as many model layers as needed to 95 reduce the cloud optical thickness (COT) in the infrared region by three units. The infrared COT was calculated using the wavelength band of 2.38 μm-4.00 μm to match the MODIS retrievals done at the 2.13 μm and 3.7 μm channels. Both CER and CDNC were calculated as extinction coefficient $b_{\mathrm{ext}}$ weighted average values that consider all model layers in the top part of the cloud (Equations (S.1) and (S.2)). For the sake of simplicity we refer to this region as the *extended cloud top*. COT was calculated for the visible wavelength band of 0.25 μm-0.69 μm as a surrogate of MODIS retrievals done at 0.66 μm. More 100 details on the sampling methodology can be found in section S2 of the supporting information.





Values for CDNC ($N_{\mathrm{d}}$) were calculated with CER ($r_{\mathrm{e}}$) and COT ($\tau_{\mathrm{c}}$) obtained for the extended cloud top in each cloud column from the following equation

$$N_{\mathrm{d}} = \frac{\sqrt{5}}{2\pi k}\left(\frac{f_{\mathrm{ad}}c_{\mathrm{w}}\tau_{\mathrm{c}}}{Q_{\mathrm{ext}}\rho_{\mathrm{w}}r_{\mathrm{e}}^5}\right)^{\frac{1}{2}}, \tag{1}$$

where $k$ is the the relation of volume mean radius and effective radius of the droplet size distribution, $f_{\mathrm{ad}}$ is the adiabaticity

factor, $c_{\mathrm{w}}$ is the rate of increase of liquid water content with height in a moist adiabatically ascending air parcel, $Q_{\mathrm{ext}}$ is the Mie extinction efficiency, and $\rho_{\mathrm{w}}$ is the density of water (Grosvenor et al., 2018b). The parameters $k$, $f_{\mathrm{ad}}$, $c_{\mathrm{w}}$, $Q_{\mathrm{ext}}$, $\tau_{\mathrm{c}}$, and $r_{\mathrm{e}}$ were diagnosed from the UCLALES-SALSA model.

The cloud parameters $k$, $f_{\mathrm{ad}}$, $c_{\mathrm{w}}$, $Q_{\mathrm{ext}}$ and $\rho_{\mathrm{w}}$ can be assumed to be constant and denoted by $\alpha$ for which an often used value for marine stratiform clouds is $1.37 \times 10^{-5}\,\mathrm{m}^{-\frac{1}{2}}$ (Quaas et al., 2006; Grosvenor et al., 2018b; Gryspeerdt et al., 2022;

Arola et al., 2022). Consequently, CDNC values can be obtained from

$$N_{\mathrm{d}} = \alpha\tau_{\mathrm{c}}^{\frac{1}{2}}r_{\mathrm{e}}^{-\frac{5}{2}}. \tag{2}$$

LWP values were calculated with the following equation (Wood, 2006)

$$\mathrm{LWP} = 5/9\rho_w r_{\mathrm{e}}\tau_{\mathrm{c}}. \tag{3}$$

In the analysis, we filtered the data so that we only considered cloudy columns where $\tau > 4$ and $4\,\mathrm{\mu m} < r_{\mathrm{e}} < 15\,\mathrm{\mu m}$ similar to

Gryspeerdt et al. (2019) and Arola et al. (2022). An example of cloud field properties can be seen in Figure S3.

## 2.2    RESULTS

### 2.2.1    The effect of cloud internal variability on retrieved CDNC and LWP

First, to get an indication on how the cloud cell level variability affects the satellite retrieved LWP adjustment, we compared model predicted CDNC and LWP values with CDNC and LWP values calculated using Equations (1)-(3). We carried out an

ensemble of UCLALES-SALSA simulations, varying the conditions for cloud formation and the number concentrations of aerosol particles, and then analysed the simulated CDNC and LWP values with the approaches detailed below.

As an example, Figure 2 shows the cloud droplet number concentration over the model domain of a simulation where the model was initialized with total aerosol number concentration of $300\,\mathrm{cm}^{-3}$. The leftmost panel in Figure 2 represents CDNC values diagnosed directly from the UCLALES-SALSA model. In the middle panel, CDNC was calculated from Equation (1),

using LES simulated values for $k$, $f_{\mathrm{ad}}$, $c_{\mathrm{w}}$, $Q_{\mathrm{ext}}$, $\tau_{\mathrm{c}}$, and $r_{\mathrm{e}}$ in the equation. In the rightmost panel CDNC was calculated from Equation (2) assuming constant $\alpha$ of $1.37 \times 10^{-5}\,\mathrm{m}^{-\frac{1}{2}}$, and using simulated $\tau_{\mathrm{c}}$ and $r_{\mathrm{e}}$. The probability distributions of CDNC and LWP data are shown at the top and to the right of the coordinate frame, respectively in Figure 3. From these distributions we can see that LES derived data is skewed towards higher CDNC values while the LWP probability distributions look very similar for both LES and satellite equations. The sharp cut off of data at low LWP and CDNC values are caused by filtering

of the data to include only values where $\tau > 4$ and $4\,\mathrm{\mu m} < r_{\mathrm{e}} < 15\,\mathrm{\mu m}$. Earlier studies on satellite data limit this filtering to



**Figure 2.** CDNC at the cloud top a) from the direct output of UCLALES-SALSA, b) calculated using Equation (1) with UCLALES-SALSA simulated values for all parameters, c) using Equation (2), d) relative biases in CDNC between UCLALES-SALSA and Equation (1), e) relative biases in CDNC between UCLALES-SALSA and Equation (2).

CDNC but not to LWP (Gryspeerdt et al., 2019). However, due to doing pixel-by-pixel analysis for CDNC-LWP correlation, both CDNC and LWP data are filtered here.





The leftmost panel shows a closed cell type structure in the cloud with lower values for CDNC at the boundaries of the cells. Note that the simulated cloud in Figure 2 is fully overcast, also at the cloud cell edges. Comparing Figures 2a and b, we

can see that when all the parameters in Equation (1) are from LES simulations, CDNC corresponds quite well with the model values, showing similar structure although overestimating the CDNC throughout the model domain. However, the averaging of satellite data will mitigate this since spatial aggregation of the data will reduce the maximum CDNC values (see the differences between Figures S9 and S10 in Supporting Information) making the CDNC distribution more narrow (Figure S11). This is also in line with observations where aircraft and satellite observed CDNC are compared (Gryspeerdt et al., 2022).

Comparing Figures 2a and c, we can see that when we use a fixed value for $\alpha$, the satellite equation exhibits inverse behaviour at cloud cell boundaries compared to the direct output of the model, i.e., CDNC increases towards the boundaries of the cloud cells. This indicates that the assumptions of e.g., adiabaticity does not hold at the cloud cell boundaries. This is also in line with a previous study by Feingold et al. (2022). Biases in LWP also occur at cloud cell boundaries (Figure S5).

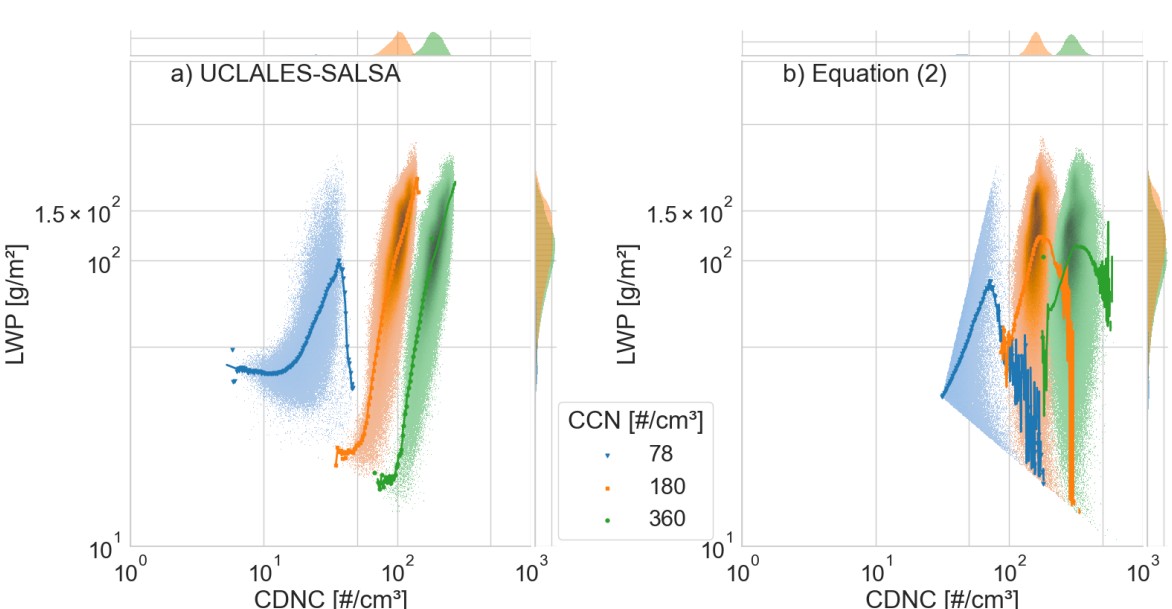

**Figure 3.** Joint and marginal histograms for LWP and CDNC values using a) UCLALES-SALSA and b) Equation (2) at a time instance of 6 hours. Simulations are colour coded according to CCN concentrations used in the model initialization. The intensity of colour in joint histograms increases when the probability increases. The probability is represented as a density function calculated as counts/sum(counts)/bin area. Continuous lines indicate the arithmetic mean.

To see how the discrepancy between the cloud properties diagnosed directly from UCLALES-SALSA and using Equation

(3) translate to differences in the correlation between CDNC and LWP, we calculated this relation for all ensemble members. Figure 3 illustrates LWP as a function of CDNC for direct model input and calculated using Eqs (2) and (3) for the three





different initial CCN concentrations at 6 hours into the simulation. For all initial CCN concentrations, the direct model output indicates an almost linear correlation in the log-log scale within single cloud scenes. The lowest CCN case exhibits a drop in LWP at highest CDNC values. However, the amount of data points is very low at highest CDNC values. In contrast, when
using Equations (2) and (3) the variability and the unphysical behaviour in $N_d$ at the boundaries of cloud cells yield in curves which reach a local maximum and for higher CDNC values exhibit a downward slope with increasing CDNC. In addition, CDNC values have a clear high bias. In this case, satellite derived CDNC values are at least two times higher than the direct LES values. Although the spatial resolution in LES is much higher than in satellite data and it has been shown that spatial and temporal averaging affects the CCN-LWP correlation (Rosenfeld et al., 2023), this behaviour is similar to what is seen in
satellite data and what was also demonstrated with synthetic data by Arola et al. (2022). This same behaviour was seen in all of the ensemble members over all analysed time instances and all CCN concentrations (Figures S6-S8).

Within individual ensemble members, the cloud internal variability contributes to the CDNC-LWP correlation and cannot be considered to be an aerosol effect on clouds. In addition, in the analysis of satellite data, it is a common practice to avoid non-adiabatic clouds (e.g. Grosvenor et al., 2018b), so that issues in satellite retrieved cloud properties at, for example, cloud
cell edges can be avoided. Previous studies have shown that selecting adiabatic pixels in a model and satellite analysis bring their results closer to each other (Dipu et al., 2022).

### 2.2.2 The effect of combining different aerosol and time instances on CCN-LWP correlation

In addition to internal variability within clouds due to the dynamics affecting the cloud structure, cloud scenes can include clouds at different phases, e.g., transitioning between closed cell structure to open cell structure. Cloud scenes can also include
large scale variability in the cloud geometric thickness and water content which are due to differences in meteorological conditions rather than caused by differences in aerosol concentration. To get an indication on how such variability affects the correlation between LWP and CDNC, we analysed simulated cloud scenes at different points in time (2 h, 6h, and 10 h). During the simulation, the closed cell structure transformed to an open cell structure for the case with the lowest initial aerosol load, and for the cases of higher aerosol load the size of convective cells increased.

Figure 4 shows the correlation between CDNC and LWP from direct model output and calculated using Equations (2) and (3) when all the cloud scenes are aggregated. Figure 4a shows that the model produces an overall positive correlation while satellite equations produce a similar shape correlation as shown in Figure 3b for one ensemble member, only spreading over a wider CDNC range due to the variability in CDNC concentrations. The direct model output resembles the shape of the correlation between CDNC and LWP simulated by the ICON model in Fons et al. (2024), while the satellite equation exhibits
a decrease in LWP at CDNC values higher than $300\,\mathrm{cm}^{-3}$.

To further investigate the differences between the LES model and satellite equations, we compared the mean LWPs of the LES domain using both direct output and satellite equation derived LWP for different initial aerosol concentrations. Figure 5 shows the LES domain mean LWP at three different time instances into the simulation for three different runs as a function of the initial CCN concentration. Solid lines denote the mean LWP in the domain and the shading indicates the standard deviation
in the data.



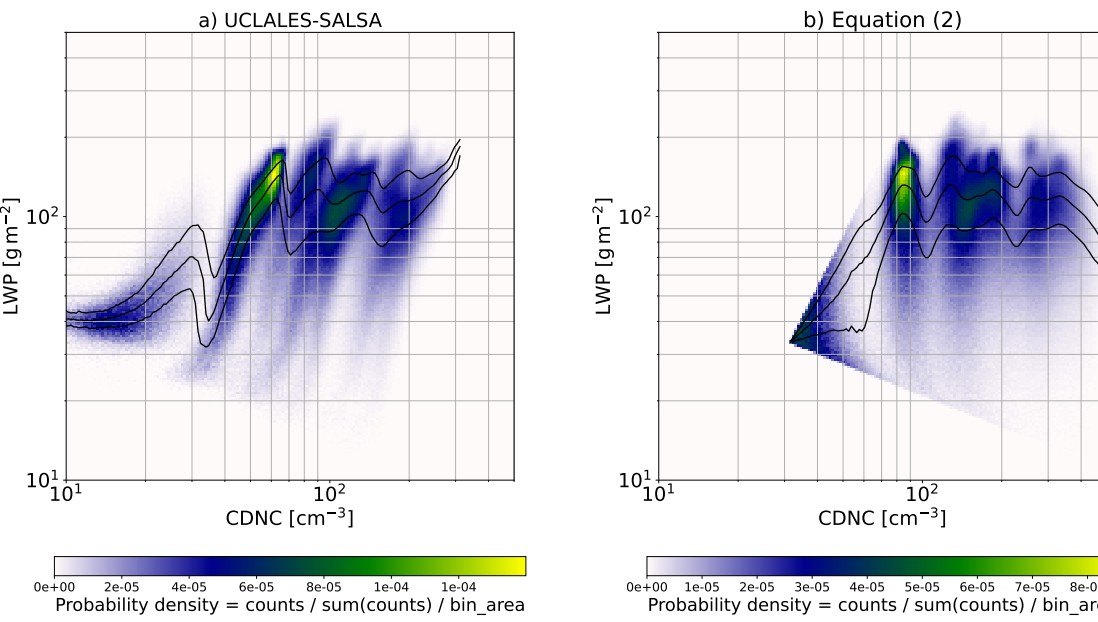

**Figure 4.** Joint histogram of LWP as a function of CDNC a) from the direct output of UCLALES-SALSA, b) calculated using Equations (2) and (3) assuming a constant $\alpha$. Black continuous lines indicate the $25^{th}$, $50^{th}$ and $75^{th}$ percentiles of LWP per bin. The color scale indicate the probability density calculated as counts/sum(counts)/bin area.

Early into the simulation, we expect the simulated clouds to be close to a very similar phase of the cloud cycle and thus the difference in LWP between the different aerosol cases comes mainly from the differences in CCN concentrations. During the first two hours into the simulation there has not been enough time for precipitation to develop, and thus the LWP decreases slightly as a function of CCN concentration. Previous LES studies of the same DYCOMS case have produced qualitatively similar results (Ackerman et al., 2009; Bulatovic et al., 2019). Later in the simulation, precipitation is initiated with the lowest CCN concentration, and a typical shape of CCN-LWP correlation is reached where LWP first increases with CCN, and then decreases due to increased entrainment rate (Ackerman et al., 2009; Bulatovic et al., 2019). However, the difference in LWP between CCN concentrations of $180 \mathrm{cm}^{-3}$ and $360 \mathrm{cm}^{-3}$ does not change significantly from 6 to 10 hours into the simulation, which is opposite to the findings in Glassmeier et al. (2021), and can be characteristics to DYCOMS2 input profile with quite small moisture inversion. The figure also illustrates, that although we also included non-adiabatic model grid points in calculations where Equations (2) and (3) are used, changes in LWP with increasing CCN are strikingly similar to those diagnosed from the LES model. There is a slight bias in satellite equation derived LWP (Figure S5) in all the cases, but the relative changes correspond well with LES diagnosed relative changes in LWP.

This analysis indicates that the non-adiabaticity of the cloud cell edges does not contribute significantly to the "inverted v" shaped correlation between CDNC and LWP seen in satellite data. Although there are issues in using Equations (2)-(3), coarse resolution of satellite data will reduce these issues significantly. Due to its coarse resolution, satellite data can include both





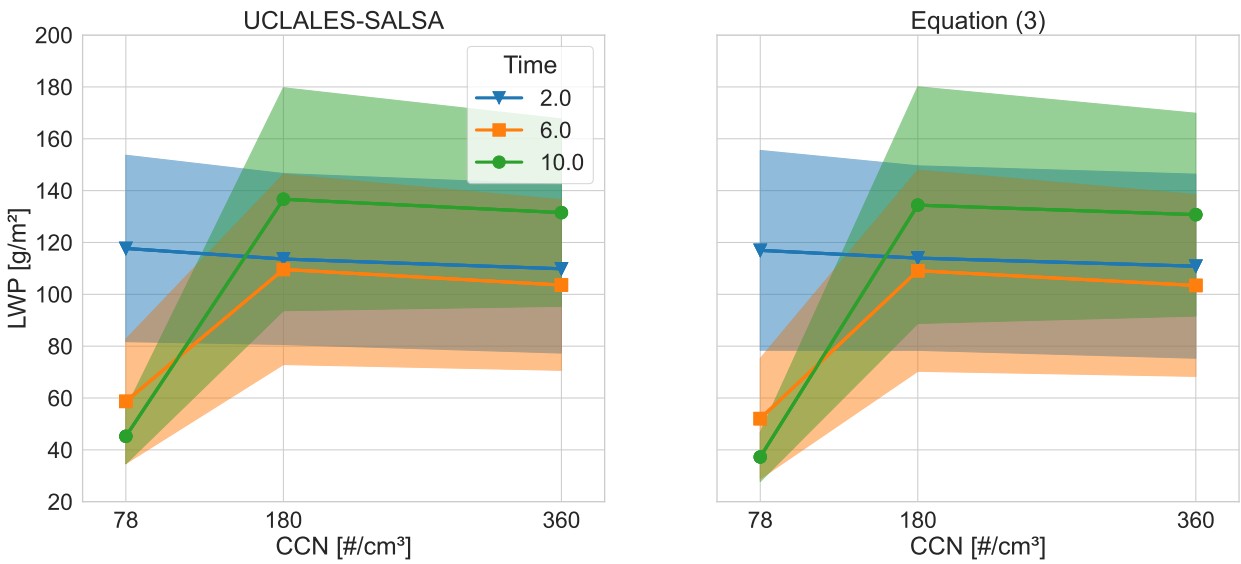

**Figure 5.** LWP as a function of CCN from the direct output of UCLALES-SALSA and calculated using Equation (3). Time instances are color coded. Shaded areas indicate the spread of values in terms of the standard deviation.

cloud cell centers and edges and could therefore, introduce bias in the retrieved LWP values. However, based on Figure 5, in our simulated cases this aggregation of different cloud structures does not affect the derived response in LWP to changes in CCN. We also tested this by using spatial averaging of $1.425\,\mathrm{km}$ by $1.425\,\mathrm{km}$ to correspond to the spatial resolution of satellite
data. Since radiances are directly proportional to cloud optical thickness, we use COT values in cloudy columns as a weighting factor to perform horizontal averaging operations along subdomains. Figures S9 and S10 in Supporting Information illustrate that spatially averaged data shows very similar cloud field properties with less frequent large CDNC relative deviations because averaging reduces the variability in $r_\mathrm{e}$ (Figure S11). The CCN-LWP shaped correlation for different time instances and CCN scenarios lack of the inverted v-shape (Figures S12, S13, and S14).

### 2.2.3   The effect of satellite instrument uncertainty or variability on retrieved CDNC and LWP

In addition to variability in cloud properties that originate from variability in aerosol concentrations and meteorological conditions, satellite instruments also include uncertainty originating from instrument noise as well as three-dimensional radiative effects. Such variability or noise will further affect the correlation between CDNC and LWP. Here we repeated the analysis combining all analysed cases and adding $20\,\%$ variability in both $\tau_\mathrm{c}$ and $r_\mathrm{e}$ then calculating CDNC and LWP from Equations
(2) and (3). The level of variability was chosen to be in line with those used in Arola et al. (2022).





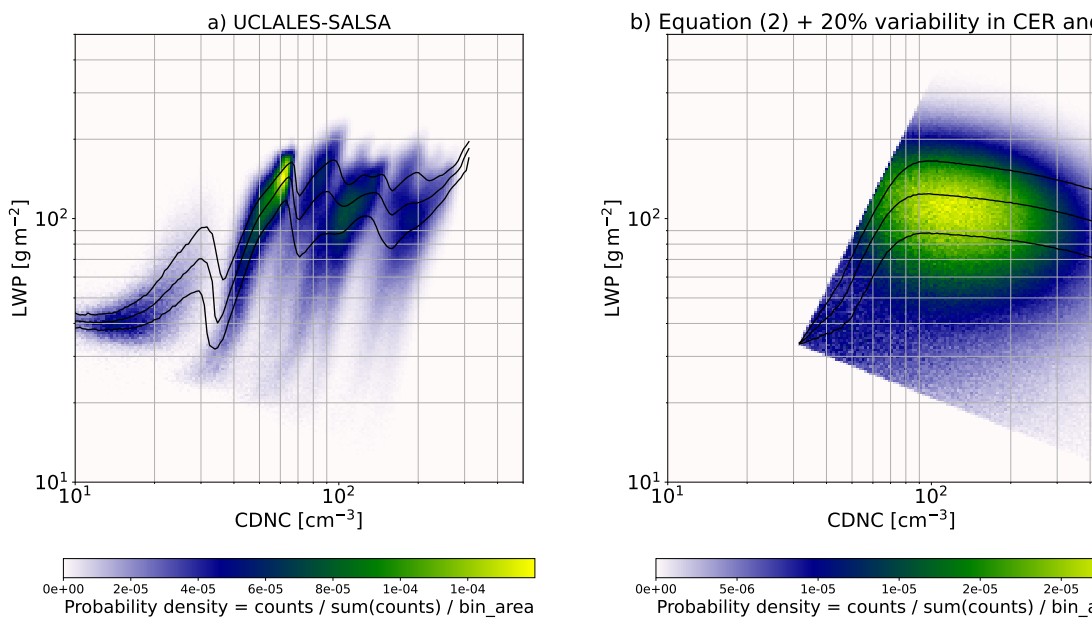

**Figure 6.** Joint histogram of LWP as a function of CDNC calculated using Equations (2) and (3) assuming a constant $\alpha$ and including 20% variability in $\tau_c$ and $r_e$. Black continuous lines indicate the 25th, 50th and 75th percentiles of LWP per bin. The color scale indicate the probability density calculated as counts/sum(counts)/bin area.

Figure 6 shows that adding noise produces a CCN-LWP correlation which increases, reaches a local maximum and is followed by a decreasing CCN-LWP correlation. However, the additional noise does not affect the correlation between CDNC and LWP compared to Figure 4b. Both Figures 3b) and 4b) exhibit a similar "inverted v" behaviour of the same magnitude. At high CDNC values the negative correlation becomes less pronounced due to the variability in x-scale, i.e., increased variability in CDNC.

## 3 Conclusions

Our LES simulations show that variability in cloud dynamics will bias satellite derived correlation between CDNC and LWP similar to Arola et al. (2022). The root cause for this is that variability in cloud effective radius causes stronger positive bias in cloud droplet number concentration when using the retrieval equation (2). Although our LES simulations include a detailed description of aerosol-cloud interactions, they do not consider the following potential error sources: 3D radiative effects in broken cloud fields, viewing geometry effects on the penetration depth bias, cloud heterogeneity at regional scale, and changes in surface reflectivity induced by changes in cloud coverage (Grosvenor et al., 2018a). Nonetheless, we could not find evidence in our model results to validate the persistent negative LWP adjustment predicted by satellite-equations for stratocumulus clouds affected by aerosol perturbations.





Cloud cell level variability in cloud effective radius and cloud optical thickness caused significantly different response in LWP with respect to changes in CDNC within individual simulations with different aerosol loads. However, when comparing the direct output of LWP from LES and those derived using Equation (3) for different CCN values, both show remarkably similar response between CCN and LWP. This indicates that although adiabaticity assumption near cloud edges causes error in CDNC and LWP, on the average Equation (3) derived LWP corresponds well with averaged LWP diagnosed from the LES

output (see Figure 5). Furthermore, the relatively coarse resolution of satellite retrievals mitigates the impact of cloud cell level variability through averaging. However, CDNC is overestimated when using Equation (2) especially at the lower LWP values and this highlights the need to obtain a good constraint for CCN instead of using satellite derived CDNC as a proxy for CCN. Based on this, determining LWP requires careful selection of clouds, estimating the mean LWP for different aerosol loads, e.g., over regions where there has been a clear change in aerosol emissions. Since this study focuses only on one cloud

case, analysis could be extended to cover wider variability in cloud conditions using for example large scale aerosol-aware high resolution climate models, which would capture also the mesoscale variability. The "inverted v" shape functionality of LWP adjustments is seen in simulations of GCMs of current generation and could be introducing confounding effects into the effective radiative forcing of aerosol-cloud-interactions (ERFaci) (Mülmenstädt et al., 2024). This clearly highlights that the behaviour of cloud water in response to changes in aerosol remains an open question and current knowledge does not support

modifying the climate model cloud schemes to produce the "inverted v" behaviour.

*Code availability.* Large-eddy-simulations were performed with UCLALES-SALSA (DEV branch version November, 2023) available from https://github.com/UCLALES-SALSA/UCLALES-SALSA/tree/DEV. Input files used to initialise the model can be built as it is shown in section S1 of the supporting information.

*Data availability.* Datasets of cloud properties derived from simulations and equations for satellite retrievals used are available at

https://fmi.b2share.csc.fi/records/8fc77f2c6a8a4deab3de2efd46683010 (Kokkola et al., 2024)

*Supplement.* There is supplementary information available for this study including:

1. Model initial settings

2. Conditional sampling of modeled cloud properties

3. Surrogates of satellite-retrievals for CDNC and $r_\mathrm{e}$

4. Liquid water path susceptibility to changes in CDNC

5. Spatial aggregation of cloud properties

6. Spatial aggregation effects on liquid water path susceptibility to changes in CDNC





*Author contributions.* Conceptualization: HK, AA, AP, SC, JT, SR, TM, formal analysis: HK, AP, JT, SC, funding acquisition: HK, SR, TM, investigation: HK, AP, JT, SC, methodolology: HK, AA, AL, AP, JT, PK, SC, SR, THV, software: HK, JT, SR, SC, supervision: HK, AA,
SR, validation: HK, SC, visualization: HK, SC, writing - original draft preparation: HK, SC, SR, writing - review and editing: HK, AA, AL, AP, JT, SC, SR, THV, TM, EG

*Competing interests.* Authors declare that no competing interests are present.

*Acknowledgements.* This project has received funding from Horizon Europe programme under Grant Agreement No 101137680 via project CERTAINTY (Cloud-aERosol inTeractions & their impActs IN The earth sYstem), Marie Skłodowska-Curie Actions (MSCA) of the Euro-
pean Union (EU) under Grant Agreement No 101072354, Horizon 2020 programme under Grant Agreement No 821205 via project FORCeS (Constrained aerosol forcing for improved climate projections), Research Council of Finland grants No 337549 (Atmosphere and Climate Competence Centre, ACCC) and No 339885 BSOA-BORE (Are Biogenic Secondary Organic Aerosols Climatically Significant in the Boreal Region?) and a Royal Society University Research Fellowship (URF/R1/191602)



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
