# Peer review of "Model analysis of biases in satellite diagnosed aerosol effect on cloud liquid water path"

_EGUsphere, 2024_

## Author Comment (AC1)

**Authors' Response to Reviews of**

**Model analysis of biases in satellite diagnosed aerosol effect on cloud liquid water path**

Harri Kokkola, Juha Tonttila, Silvia Calderón, Sami Romakkaniemi, Antti Lipponen, Aapo Peräkorpi, Tero Mielonen, Edward Gryspeerdt, Timo H. Virtanen, Pekka Kolmonen, and Antti Arola
*EGUsphere,* 10.5194/egusphere-2024-1964, 2024
* * *
**RC:** *Reviewers' Comment*,    AR: Authors' Response,    ☐ Manuscript Text

**Reviewer #1**

https://doi.org/10.5194/egusphere-2024-1964-RC1

**General comment**

**RC:** The manuscript investigates the biases in satellite retrieval of liquid water path (LWP) and cloud condensation nuclei (CDNC) using LES simulations configured to DYCOMS-II RF02. The authors compare the correlations of the two variables from three sources: direct model outputs, retrievals from equations using LES simulated values for the parameters, and retrievals from equations assuming constant parameters. The authors find that CDNC retrievals assuming constant parameters tend to increase CDNC at the cloud boundaries. They conclude that satellite-derived CDNC shows a significant positive bias, but the correlation between LWP and CDNC was very similar between the methods. The authors also find that the instrumental noise in satellite retrievals do not affect the correlation between CDNC and LWP. The topic of this manuscript is suitable for publishing in ACP. However, the manuscript is poorly articulated, and the results are not effectively presented. The confusing variable labels in the figures further contribute to the overall lack of clarity. The current version isn't ready for publication but may be considered after some revisions. I would like the authors to respond to and address my comments. Details of my comments are as follows: Recommendation: Major revisions

**AR:** We deeply appreciate your comments, questions and suggestions. We will proceed to resolve each one of them. In some cases, we have added subsections to address each item.

**Major comments**

**RC:** **1.** I don't find the figures to provide sufficient support for the authors' arguments in the text. I suggest adding more figures to better substantiate the arguments. Please see the detailed comments below.

**AR:** We have addressed your comments and modified the figures accordingly. A more detailed explanations of modifications will be provided later in this reply.

**RC:** **2.** The axis variables are not clearly labeled. The authors need to clearly label the LWP and CDNC in Figs. 3-6, indicating whether they are direct model outputs or computed from equations 1-3, and whether they are pixel-level data or domain-average data. An easy way to address this is by using subscripts, e.g., LWPtrue, LWPeq3, CDNCeq2, etc. Use overhead bars if the variables are domain average values. The titles of Figs. 3b, 4b, and 6b are confusing because the LWP in these panels is computed from equation 3, not equation 2. I

suggest using a title like "Computed LWP and CDNC" or something similar.

**AR:** This is indeed a very good suggestion. We have changed the variable names according to it.

**Detailed comments**

**RC:** **1.** caption: "the calculated CDNC". Is it calculated from eq. 1 or 2? "the retrieved LWP". Is it calculated from eq. 3? Please also specify the sizes of the squares on the right panels.

**AR:** CDNC is calculated from Eq. (2) and we have modified the text to read CDNC calculated according to **?**. The small squares are approximately 30 km in size, while the width of the larger rectangle is approximately 500 km. We have added this information also in the figure caption.

> Cloud properties of a stratocumulus cloud deck west of Peru and Chile over South Pacific on Aug 30th, 2003. The upper left panel shows the  CDNC calculated according to Quaas et al. (2006) and lower left panel shows the retrieved LWP from Moderate Resolution Imaging Spectroradiometer (MODIS) Level-2 (L2) Collection 6.1 **?**. Right panels show a magnification of the structure of a cloud cell within the cloud field denoting the cloud effective radius and the cloud reflectance for the corresponding cloud cell. Small squares are approximately $3\,km \times 3\,km$ and large rectangles are approx. $500\,m \times 6\,km$.

**RC:** **2.** Lines 35-38. The authors argue that the cloud top effective radius and liquid water path change with cloud top pressure. However, cloud top pressure is not shown in the paper or supplementary material. Please provide a figure of cloud top pressure to support this argument.

**AR:** Since this is introductory text and not results of this study, we have not included the figure in the manuscript or in a supplement. Below is the figure of MODIS AQUA cloud top pressure for Aug 30th, 2003.

[Figure]

Figure R1: Cloud top pressure for MODIS AQUA

**RC:** **3.** Line 41. Replace "liquid water content" with "liquid water path" as the former is not shown in Fig. 1.

**AR:** Thank you. The manuscript reflects the replacement.

> The decrease in retrieved cloud effective radius results from the entrainment mixing at the cloud top and downdrafts in the cloud cell boundaries which both reduce the liquid water path.

**RC:** **4.** Lines 41-43. I don't see how this sentence connects to the previous one. Please revise the paragraph for better flow.

**AR:** Paragraphs were rephrased as follows

> The decrease in retrieved cloud effective radius results from the entrainment mixing at the cloud top and downdrafts in the cloud cell boundaries which both reduce the liquid water path. At the cloud cell edges this is in conflict with the assumptions made in the calculation of CDNC.Calculation of CDNC based on the effective radius, and assuming constant sub-adiabaticity, would lead to increased CDNC at the cell boundaries (see Equation (2) in Section 2). In addition to actual variability in physical properties of clouds, satellite retrievals include uncertainties and instrument noise causing another potential source of bias in the satellite-derived correlation between CDNC and LWP.

**RC:** **5.** Lines 126-132. Since the section predominantly centers on Fig. 2, the brief reference to Fig. 3 interrupts the flow. To improve continuity, I suggest removing the discussion of Fig. 3.

**AR:** We have moved the discussion of Figure 3 towards the end of this subsection where Figure 3 is also otherwise discussed.

**RC:** **6.** Line 133. "The leftmost panel shows a closed cell type structure in the cloud". It's difficult to discern cloud structure in Fig. 2. Please include a snapshot of LWP to clarify.

**AR:** A snapshot of LWP is shown in Supplementary Material, Figure S3.

**RC:** **7.** In Figure 3 caption: "Simulations are colour coded according to CCN concentrations used in the model initialization". Please provide a legend in the figure to reflect this.

**AR:** Legend of Figure 3 was improved to help identifying simulation scenarios with different CCN background conditions.

**RC:** **8.** Lines 136-143. I think the paper would benefit from a separate figure comparing CDNC from model output and equations 1 and 2 to support the arguments in the text. I suggest adding a scatter plot with the true CDNC from the LES on the x-axis and the computed CDNC on the y-axis. Mark the domain-average CDNCs in the plot. It would be helpful if the authors could overlay the results of aggregation so that readers do not need to refer to the Supporting Information for details. The authors might consider coloring the scatter plot with LWP values to illustrate the bias of computed CDNC in relation to cloud structure. Similar plots can be made for LWP from model output and equation 3.

**AR:** The comparison of modeled and satellite-retrievals CDNC distributions was shown using marginal histograms in Figure 3 and Figures S6-S8 and S12-S14. We acknowledge that the size ratio between the joint and marginal histogram was not optimal and it is difficult to compare CDNC distributions. We have the correspondent figures in the main manuscript and supplement. Your suggestion is very good indeed. However, each time instance for a single simulation scenario comprises millions of points and it is very difficult to visualize trends due to overlapping layers caused by data variability.Nonetheless, we have addressed your comment and here we show a graphical comparison of CDNC distributions. Values for probability, overlapping index and correlation coefficient correspond to the entire dataset for a single time and initial aerosol loading. Scatter plots, instead, reflect random samples equivalent to 1% of the total data, approximately. CDNC distributions of model outputs and satellite-retrieval equations in a simulation initialized with a CCN loading of 360 $\mathrm{cm}^{-3}$ at the time instant

of 10 h are compared in Figure R2 for the high resolution case (75 m × 75 m) and in Figure R3 for the low resolution case (1425 m × 1425 m) obtained from spatial aggregation.

[Figure]

Figure R2: Comparison of distributions for cloud droplet number concentration (CDNC) obtained at high resolution (75 m × 75 m) from model outputs and satellite-retrieval equations in a simulation initialized with a CCN loading of $360\,\mathrm{cm}^{-3}$ at the time instant of 10 h a) Histograms of CDNC distribution indicating the overlapping index value (OVL) (i.e. If OVL=1 distributions are equivalent to each other) b) Scatter plot using LWP in the color scale and CER in $\mu$m for marker size c) Scatter plot using the adiabatic factor in the color scale and CER in $\mu$m for marker size. In the scatter plots, we have indicated linear correlation coefficient values ($p<0.05$) and added continuous black lines of perfect correlation as a visual guide. Mean values are highlighted with black edges keeping the variable color scale. For both satellite equations, larger biases correspond to thinner and subadiabatic columns with smaller droplet effective radius, conditions that are likely to happen in cloud edges. Histograms for CDNC-satellite values from Equation (2) show lower overlapping index as well as more frequent and higher positive deviations. Despite having a more robust approach that considers deviations from the adiabatic liquid water path as well as changes in the droplet distribution breadth, CDNC-satellite values from Equation (1) are still much higher than those from the model.

[Figure]

Figure R3: Comparison of distributions for cloud droplet number concentration (CDNC) obtained at low resolution (1425 m × 1425 m) from model outputs and satellite-retrieval equations in a simulation initialized with a CCN loading of 360 $cm^{-3}$ at the time instant of 10 h. a) Histograms of CDNC distribution indicating the overlapping index value (OVL) (i.e. If OVL=1 distributions are equivalent to each other) b) Scatter plot using LWP in the color scale and CER in $\mu$m for marker size c) Scatter plot using the adiabatic factor in the color scale and CER in $\mu$m for marker size. In scatter plots, we have indicated linear correlation coefficient values (p<0.05) and added continuous black lines of perfect correlation as a visual guide. Mean values are highlighted with black edges keeping the variable color scale. After spatial aggregation using COT as a weighting factor, CDNC distributions become more symmetric and less spread out around the mean which in turn results in a reduction of the overlapping index between modeled and satellite-retrieval distributions. Although the aggregated dataset have a much lower influence of model columns with thinner sub-adiabatic clouds with smaller CER values, CDNC satellite-retrievals are still higher and linearly proportional to modeled ones (i.e. correlation coefficients in Figure R3 are larger than 0.5) confirming the systematic deviation.

[Figure]

Figure R4: Joint and marginal histograms for LWP and CDNC values using a) UCLALES-SALSA and b) Equation (2) at a time instance of 6 hours. Simulations are colour coded according to CCN concentrations used in the model initialization. The intensity of colour in joint histograms increases when the probability increases. The probability is represented as a density function calculated as counts/sum(counts)/bin area. Continuous lines indicate the arithmetic mean.

**RC:** **9.** Line 142. Please list all possible assumption biases here instead of using "e.g".

**AR:** The main manuscript was modified to include possible biases.

> Biases in LWP also occur differently across cloudy areas (Figure S5).
>
> Cloud cell boundaries tend to have low biased LWP values while cloud cell centers are biased high. In cloud cell boundaries processes such as entrainment and lateral mixing leads to sub-adiabaticity. Since these sources of variability are not considered in the formulation of satellite retrieval equations, there are important deviations from the assumptions of vertically constant values for droplet number concentration, droplet size distribution breadth and adiabaticity.

**RC:** **10.** Line 152. "Satellite derived CDNC values are at least two times higher than the direct LES values". Do the authors have any idea what might be causing this?

**AR:** Satellite derived CDNC values are calculated using direct LES values of cloud effective radius (CER) and cloud optical thickness (COT) that correspond to the expected optical penetration depth. However, they are positively biased due to the non-fulfilment of the underlying assumptions in the pseudo-adiabatic cloud model (i.e. vertically constant values for droplet number concentration, droplet size distribution breadth, adiabaticity). These criteria are hardly satisfied in thin cloud layers such as those observed in cloud edges.

> In addition, CDNC values have a clear high bias. In this case, satellite derived CDNC values are at least two times higher than the direct LES values. Values are positively biased due to the assumption of vertically uniform cloud

columns which is not valid in thin cloud layers such as those observed in cloud edges.

**RC:** **11.** Line 157. "Within individual ensemble members, the cloud internal variability contributes to the CDNC-LWP correlation and cannot be considered to be an aerosol effect on clouds". Zhou and Feingold, (2023) has reached a similar conclusion. (Zhou, X., & Feingold, G. (2023). Impacts of mesoscale cloud organization on aerosol-induced cloud water adjustment and cloud brightness. Geophysical Research Letters, 50(13), e2023GL103417.)

**AR:** We have added the reference to Zhou and Feingold in the revised manuscript.

> Within individual ensemble members, the cloud internal variability contributes to the CDNC-LWP correlation and cannot be considered to be an aerosol effect on clouds, also shown by **?**.

**RC:** **12.** Line 177. To improve the connection with the subsequent discussion, please make it clear that Fig. 5 represents a proxy for satellite aggregation.

**AR:** The text in the manuscript has been modified as follows:

> Figure 5 represents a proxy for satellite aggregation. It shows the LES domain mean LWP at three different time instances into the simulation for three different runs as a function of the initial CCN concentration. Solid lines denote the mean LWP in the domain and the shading indicates the standard deviation in the data.

**RC:** **13.** Fig. 6a is identical to Fig. 4a. Replotting it is unnecessary.

**AR:** This is correct and we have removed Figure 4a as suggested.

---

## Author Comment (AC2)

**Authors' Response to Reviews of**

**Model analysis of biases in satellite diagnosed aerosol effect on cloud liquid water path**

Harri Kokkola, Juha Tonttila, Silvia Calderón, Sami Romakkaniemi, Antti Lipponen, Aapo Peräkorpi, Tero Mielonen, Edward Gryspeerdt, Timo H. Virtanen, Pekka Kolmonen, and Antti Arola

*EGUsphere,* `10.5194/egusphere-2024-1964,` 2024
* * *
**RC:** *Reviewers' Comment*,    AR: Authors' Response,    ☐ Manuscript Text

**Reviewer #2**

`https://doi.org/10.5194/egusphere-2024-1964-RC2`

**Overview**

**RC:** The joint relationship between LWP and cloud droplet number concentration (CDNC) is commonly used to infer the LWP adjustment to CDNC changes resulting from changes in CCN. This relationship has been assessed in satellite retrievals, but such retrievals depend on key assumptions in order to estimate CDNC and LWP. Biases associated with these assumptions are evaluated in LES simulations of the frequently analyzed DYCOMS RF02 case. Satellite retrievals of CDNC are high biased, and if constant values are used for adiabaticity and other inputs as commonly done, CDNC increases from cloud cell cores to edges, opposite of simulation output. Allowing those variables such as adiabaticity to vary based on model output produces the proper CDNC spatial pattern, though still with a high bias. The satellite retrieval LWP is unbiased overall if averaged across cells, though it tends to be overestimated in cell cores and underestimated on cell edges. For a set CCN concentration, LWP increases approximately linearly with CDNC on a log-log scale, and combining across different CCN simulations, the negative slope of the inverted "v" LWP-CDNC relationship is not produced. If adiabatic satellite retrievals of CDNC and LWP are used, inverted "v" shaped LWP-CDNC relationships are produced. Introducing 20% uncertainty to satellite retrievals of cloud optical depth and effective radius does not alter the overall LWP-CDNC relationship. The overestimate of CDNC is particularly biased at relatively low LWP. The authors thus argue that a good constraint on CCN is required because CDNC cannot be used as a proxy in such situations for representing LWP adjustments. Overall, the study presents compelling evidence that the negative slope portion of the inverted "v" LWP-CDNC relationship may not be caused by LWP adjustments to CDNC. This is a very informative study that is thorough in its methods. Its conclusions should aid improved interpretation of satellite retrieved cloud microphysical properties that are used to infer aerosol-cloud interactions. More studies like this are needed to sufficiently understand and design proper model-observation comparisons. I don't have any major concerns with the study but have some minor comments, mostly related to clarification, that I think could improve the study if addressed.

**AR:** We deeply appreciate your comments, questions and suggestions. We will proceed to resolve each one of them. In some cases, we have added subsections to address each item.

**Comments**

**RC:** **1.** Lines 22-24: This sentence would be more informative if it explicitly stated what the counteracting physical processes and satellite retrieval challenges were with respect to the studies cited.

**AR:** This is a very good comment and to explain this more clearly, we have modified this part to read as follows:

> These mixed results have been attributed to several counteracting physical processes, for example the effects of solar heating, cloud-top mixing, and variability in moisture on LWP (Feingold et al., (2022); Gryspeerdt et al., 2022; Glassmeier et al., 2021; Zhang et al., 2024). In addition, there are temporal and spatial averaging and variability and/or noise in satellite data which cause bias in satellite based estimates of aerosol-cloud interactions (Feingold et al., 2022; Arola et al., 2022).

**RC:** **2.** Line 38: I believe "higher boundary layer" should be "boundary layer depth change".

**AR:** The manuscript was changed accordingly.

> The cloud top pressure increases from 850 hPa to 890 hPa going from west to east indicating  boundary layer depth change which might affect boundary layer dynamics and thus cloud properties.

**RC:** **3.** Line 42: State what the increased CDNC at the cell boundaries is relative too, presumably real CDNC values?

**AR:** The manuscript was changed to clarify the sentence.

> However, calculation of CDNC based on the effective radius, and assuming constant sub-adiabaticity, would lead to overestimation in retrieved CDNC values compared to the real CDNC values at the cell boundaries (see Equation (2) in Section 2).

**RC:** **4.** Line 45: Is the estimate of the aerosol effect on LWP referenced here based on joint distributions of LWP and CDNC or how is the true effect quantified from which the bias is determined? Beyond the question of whether the LWP-CDNC joint distribution is accurately retrieved, there is the question of whether it can even be used to estimate LWP adjustment when derived from Eulerian statistics, e.g., considering that a process such as the entrainment-evaporation feedback can take several hours to days to alter the LWP in response to a change in CDNC along Lagrangian trajectories (e.g., Christensen et al. 2023, 2024).

**AR:** Yes, the estimate of the aerosol effect on LWP in Arola et al., (2022) is based on the analysis of joint distributions of CDNC and LWP. Variability in cloud fields and noise in retrievals will also affect the analysis of CDNC-LWP response when using Eulerian statistics.

**RC:** **5.** Line 108: Clarify that these parameters are not necessarily constant even though they can be and often are assumed to be constant.

**AR:** We have now clarified this in the revised manuscript as follows:

> The cloud parameters $k$, $f_{\mathrm{ad}}$, $c_{\mathrm{w}}$, and $Q_{\mathrm{ext}}$ vary with time along the cloud structure. However the actual values cannot be directly derived from MODIS observations and thus they are  assumed to be constant and denoted by $\alpha$ for which an often used value for marine stratiform clouds is $1.37 \times 10^{-5}\,\mathrm{m}^{-\frac{1}{2}}$ (Quaas et al., 2006; Grosvenor

et al., 2018b; Gryspeerdt et al., 2022; Arola et al., 2022) Estimates of $f_{ad}$ could possibly be improved combining MODIS/CALIOP observations.

**RC:** **6.** Line 143: I see low biases in LWP at the cloud cell boundaries and high LWP biases in the cell centers.

**AR:** We have modified the text to read:

Biases in LWP also occur differently across cloudy areas (Figure S5).

Cloud cell boundaries tend to have low biased LWP values while cloud cell centers are biased high. In cloud cell boundaries processes such as entrainment and lateral mixing leads to sub-adiabaticity. Since these sources of variability are not considered in the formulation of satellite retrieval equations, there are important deviations from the assumptions of vertically constant values for droplet number concentration, droplet size distribution breadth and adiabaticity.

**RC:** **7.** Figure 3 and discussion about it: The positive slope of LWP with respect to CDNC is often assumed to be caused by precipitation suppression. That could be the case when combining multiple different CCN simulations, but for a single simulation, the positive slope is simply representing the horizontal structure of the cell where air moves from the high LWP, high CDNC core outward toward low LWP, low CDNC cell edges, correct? Is it worth clearly distinguishing between these 2 causes and interpretations?

**AR:** This is correct and a good point. We have added the following text to discuss this:

For an individual simulation, the positive slope between CDNC and LWP reflects the horizontal structure of the cell, where air flows from the core, characterized by high LWP and high CDNC, outward toward the cell edges with lower LWP and CDNC.

**RC:** **8.** Line 161-162: Making adiabaticity the same in a model and satellite analyses also brings CDNC-LWP relationships into better alignment (e.g., Fig. 21 in Varble et al., 2023).

**AR:** We truly appreciate that you have pointed out this reference. The following changes have been made to the main manuscript.

Previous studies have shown that selecting adiabatic pixels in a model and satellite analysis bring their results closer to each other (Dipu et al., 2022  and Varble et al. [2023] showed that removing the differences between the adiabaticity in an Earth System Model and satellite retrievals brings the observed and satellite retrieved LWP adjustment closer to each other.)

**RC:** **9.** Lines 193-194: I don't completely follow the argument here regarding subadiabatic points not contributing to the LWP-CDNC inverted "v" shape since it is referencing the relationship of LWP with CCN rather than CDNC in Figure 5. Adiabaticity will impact the CDNC calculation, but CCN is not impacted, so what allows for the connection of Figure 5 to the LWP-CDNC correlation?

**AR:** Here we mean to say that even though in the high resolution of the LES model the sub-adiabatic points contribute to the CDNC-LWP "inverted-v" shape in data for individual simulations, the spatial averaging of the model data to a resolution that corresponds closer to the satellite data dilutes this effect. This results in very similar CCN-LWP correlation between the direct model diagnosed and satellite-equation diagnosed CCN and LWP. Using CDNC instead of CCN would result in an "inverted-v" shape if many different cloud cases are combined. This highlights the need for using observed CCN instead of using CDNC as a proxy for CCN.

> This analysis indicates that when using domain averaged CCN and LWP values the non-adiabaticity of the cloud cell edges does not contribute significantly to the "inverted v" shaped correlation between CDNC and LWP seen in satellite data. Although there are issues in using Equations (2)-(3), coarse resolution of satellite data will reduce these issues significantly.

**RC:** **10.** Lines 203-204: This is true but if you combine Figures S12-14 like was done for Figures 4-5, could an inverted "v" LWP-CDNC correlation be produced?

**AR:** Yes, see Figure R1

[Figure]

Figure R1: Joint histogram for spatially aggregated LWP and CDNC values a) from the direct output of UCLALES-SALSA, b) calculated using Equations (1) and (2) assuming a constant $\alpha$. Black continuous lines indicate the 25[th], 50[th] and 75[th] percentiles of LWP per bin. The color scale indicate the probability density calculated as counts/sum(counts)/bin area.

**RC:** **11.** In the supplemental material, it mentions that a negative LWP-CDNC slope is produced when combining different times (cloud types) and CCN concentrations, which seems important, but I didn't see it highlighted in the main manuscript (although maybe I missed it or misinterpreted the supplemental text).

**AR:** We have now highlighted this point in the first sentence of Conclusions

> Our LES simulations show that variability in cloud properties when including different cloud types, CCN concentrations, and clouds in different phases of their cycle will bias satellite derived correlation between CDNC and LWP similar to Arola et al. [2022].

**RC:** **12.** Supplemental text line 80: what is the correction factor introduced?

**AR:** The adiabatic factor defined by Brenguier et al. [2000] considers that the adiabatic value of the liquid water path increases linearly with increasing altitude from zero at the cloud base to its maximum value at the cloud top being equal to $\mathrm{LWP}_{\mathrm{adiab}} = 0.5 c_{w,\mathrm{model}} H^2$ where $c_{w,\mathrm{model}}$ is the water condensational lapse rate in the

extended cloud top region, $H$ is the cloud geometrical thickness. In our study, we defined the cloud base differently as the minimum altitude at which the liquid water content is equal or higher than 0.01 g m$^{-3}$ instead of zero. To have comparable conditions at cloud base, we introduced the term $\mathrm{LWC_{model,CB}}H$ in Equation S.6

$$f_{\mathrm{ad}} = \frac{\mathrm{LWP}}{0.5c_{w,\mathrm{model}}H^2 + \mathrm{LWC_{model,CB}}H},$$

**AR:** The text in the supporting information was modified in accordance to the previous paragraph.

>
>
> The adiabatic factor defined by Brenguier et al. (2000) considers that the adiabatic value of the liquid water path increases linearly with increasing altitude from zero at the cloud base to its maximum value at the cloud top being equal to $\mathrm{LWP_{adiab}} = 0.5c_{w,\mathrm{mathrmmodel}}H^2$ where $c_{w,\mathrm{model}}$ is the water condensational lapse rate in the extended cloud top region, $H$ is the cloud geometrical thickness. In our study, we defined the cloud base differently as the minimum altitude at which the liquid water content is equal or higher than 0.01 g m$^{-3}$ instead of zero. To have comparable conditions at cloud base, we introduced the term $\mathrm{LWC_{model,CB}}H$ in Equation S.6

**RC:** **13.** Figure S5: The CER plot is saturated from the maximum value of 10 $\mu$m. Perhaps extend the CER range to show more structure. Also, the LWP Equation (3) panel uses a color scale that is not ideal for anomalies because it isn't clear where the zero crossover is. I suggest a diverging color scale or a contour of the 0 line (if not too noisy).

**AR:** Thanks. The color scale used to represent biases in LWP satellite retrievals was changed to a divergent one as it is shown in Figure R2. Changes are reflected in the new version of the supplement.

[Figure]

Figure R2: Biases between modelled and surrogate satellite-retrievals of LWP with Equation (3) in a simulation initialized with a CCN loading of $360\,\mathrm{cm^{-3}}$ at the time instant of 10 h

**RC:** **14.** Supplemental text lines 92-93: Larger biases are seen at cloud edges, but are the cores biased in the opposite direction too?

**AR:** This is correct, we have modified the text as follows:

> Nonetheless, biases in LWP between the model and Equation (3) are significantly lower remaining below ±20% with larger values at cloud edges and adiabatic cores as can be seen in Figure S5 (R3). All datasets at different simulation times show similar trends suggesting that biases are caused by processes at cloud edges related to stratocumulus dissipation (e.g. evaporative cooling during cloud top mixing or lateral mixing) which are not considered in the pseudo-adiabatic cloud model from which satellite equations are derived. Positive biases in satellite retrievals of LWP can be also expected when cloud top CER values do not reflect droplet growth fully driven by adiabatic cooling but instead correspond to super-adiabatic droplet growth after entrainment mixing (e.g., Yang et al., (2016), Zhu et al., (2019).

[Figure]

Figure R3: Biases between modelled and surrogate satellite-retrievals of LWP with Equation (3) in a simulation initialized with a CCN loading of $360\,\mathrm{cm^{-3}}$ at the time instant of 10 h in a section of the model domain shown in Figure 3.

**RC:** **15.** Figure S10: Why does the color bar have such a large range? It's difficult to see structure in the plots because of this.

**AR:** This is a good point. The color scale was improved to show better the cloud structure.

**RC:** **16.** Supplemental text lines 140-141: If using words like "extreme" and "very" here, I suggest adding numerical values to reference since interpretations of these words varies. Further, I suggest softening this sentence a bit to say that the satellite-retrieved inverted "v" can be caused by these biases rather than it is caused by them since this is based off of a single LES case.

**AR:** You are right. The following changes were made to the supplement:

This factor together with the fact that the signal from cloud edges is flatten out after spatial aggregation of cloud properties (Figure S9), support the hypothesis that the inverted-V shape in satellite-based studies  is likely related to  positive biases in satellite retrievals of CDNC at small CER values. Additional cloud modelling studies reflecting a wider palette of meteorological conditions and background aerosol loadings would be needed to offer a definitive confirmation.

**References**

A. C. Varble, P.-L. Ma, M. W. Christensen, J. Mülmenstädt, S. Tang, and J. Fast. Evaluation of liquid cloud albedo susceptibility in e3sm using coupled eastern north atlantic surface and satellite retrievals. *Atmospheric Chemistry and Physics*, 23(20):13523–13553, 2023. . URL https://acp.copernicus.org/articles/23/13523/2023/.

Antti Arola, Antti Lipponen, Pekka Kolmonen, Timo H Virtanen, Nicolas Bellouin, Daniel P Grosvenor, Edward Gryspeerdt, Johannes Quaas, and Harri Kokkola. Aerosol effects on clouds are concealed by natural cloud heterogeneity and satellite retrieval errors. *Nature Communications*, 13(1):7357, 2022.

Jean-Louis Brenguier, Hanna Pawlowska, Lothar Schüller, Rene Preusker, Jürgen Fischer, and Yves Fouquart. Radiative properties of boundary layer clouds: Droplet effective radius versus number concentration. *Journal of the Atmospheric Sciences*, 57:803–821, 2000. . URL https://journals.ametsoc.org/view/journals/atsc/57/6/1520-0469_2000_057_0803_rpoblc_2.0.co_2.xml.

---

## Author Response (AR2)

**Reviewer report:**
Review of "Model analysis of biases in satellite diagnosed aerosol effect on cloud liquid water path" Harri Kokkola et al., 2024

The authors have done an excellent job incorporating my comments. Thanks for including Figs. S15 and S16, I think the figures are intuitive and helpful. The manuscript is now cohesive and well-presented. I have only a few additional comments and questions for clarification. Recommendation: Minor revision.

1. Lines 38-39: If the authors do not show figures of cloud top pressure, then I suggest not including the detailed description (e.g., 850 hPa-890 hPa) as it lacks sufficient support and is not convincing.

- **Author reply:** We have removed the following sentences "Although some of the variability can come from changes in aerosol, over that region the cloud top effective radius and liquid water path follow the changes in cloud top pressure. The cloud top pressure increases from 850 hPa to 890 hPa going from west to east indicating {boundary layer depth change} which might affect boundary layer dynamics and thus cloud properties."

2. Lines 114-115. The sentence is duplicated.

- **Author reply:** We have removed the duplicated sentence "Estimates of $f_{ad}$ could possibly be improved combining MODIS/CALIOP observations".

3. Figs. 1-4 examine the correlation between CDNC and LWP, yet Fig. 5 highlights aggregated CCN and LWP. Would it make more sense to show aggregated CDNC instead, as it is more relevant? CCN data cannot be retrieved from satellites after all.

- **Author reply:** The reason we have CCN on the x-axis illustrates one of our main conclusions that if cloud condensation nuclei concentrations are well-defined, changes in liquid water path due to changes in aerosol can be confidently determined using satellite data. Although CCN cannot be retrieved satellite data, there are for example ways to combine in situ and satellite data to get better estimates of CCN. We have left the Figure as it is and added the following sentence to justify the figure better: "*This indicates that if cloud condensation nuclei concentrations are well-defined, changes in liquid water path due to changes in aerosol can be confidently determined using satellite data.*"